# GATO: GATES ARE NOT THE ONLY OPTION

## ABSTRACT

Recurrent Neural Networks (RNNs) facilitate prediction and generation of structured temporal data such as text and sound. However, training RNNs is hard. Vanishing gradients cause difficulties for learning long-range dependencies. Hidden states can explode for long sequences and send unbounded gradients to model parameters, even when hidden-to-hidden Jacobians are bounded. Models like the LSTM and GRU use gates to bound their hidden state, but most choices of gating functions lead to saturating gradients that contribute to, instead of alleviate, vanishing gradients. Moreover, performance of these models is not robust across random initializations. In this work, we specify desiderata for sequence models. We develop one model that satisfies them and that is capable of learning long-term dependencies, called GATO. GATO is constructed so that part of its hidden state does not have vanishing gradients, regardless of sequence length, and so that its state does not explode. We study GATO on copying and arithmetic tasks with long dependencies and on modeling intensive care unit and language data. Training GATO is more stable across random seeds and learning rates than GRUs and LSTMs.

## 1 INTRODUCTION

RNNs allow us to model structured data like text, image, and sound from domains such as art, healthcare, and environmental sciences. These models and their deep variants form the basis for tasks such as generation and prediction (Pascanu et al., 2013). Modeling joint distributions of sequences facilitates the generation of images (Oord et al., 2016), videos (Denton & Fergus, 2018), drawings (Ha & Eck, 2017), and jazz music (Huang et al., 2019). Modeling conditional distributions enables applications like predicting patient outcomes in medical centers with electronic health records (Che et al., 2018). In all of these cases, the data generation process may involve dependencies that range over hundreds of time steps (e.g. number of video frames or health over a lifetime) where early events may determine the distribution of later events (Kamiya et al., 2015; Denton & Fergus, 2018).

The basic RNN (Elman, 1990) suffers from vanishing and exploding gradient problems (Bengio et al., 1994) that cause difficulties for learning long-range dependencies. The problems stem from extreme eigenvalues of parameter matrices in the RNN. Such models can also suffer from unbounded growth or shrinkage of the hidden state over long sequence lengths. This can happen even when the hidden-to-hidden Jacobian does not vanish or explode and contributes additional difficulties for learning recurrent and decoder parameters. Careful parameterization and initialization of RNNs is crucial for their success in learning long-range dependencies using gradient-based optimization.

The majority of proposed solutions make use of RNNs with gated units, notably LSTMs (Hochreiter & Schmidhuber, 1997), GRUs (Cho et al., 2014) and Recurrent Highway Networks (RHNs) (Zilly et al., 2017). These models and deep variants have been the basis for many state-of-the-art empirical results. Gated architectures address gradient issues by bounding the hidden state and designing the recurrence to copy the state forward in time. The GRU and RHN update their state with a weighted average of bounded updates and the current state, where the weighted average in GRU is a convex combination. The LSTM uses an auxiliary cell state with linear recursive updates to copy the hidden state forward. However, gated models are difficult to optimize in practice, in part due to the saturating gradients of gating non-linearities.

An alternative to gated models is to restrict hidden transformations to be unitary (Arjovsky et al., 2016; Wisdom et al., 2016; Jing et al., 2017; Mhammedi et al., 2017). Unitary matrices can be safely raised to powers without vanishing gradients since they have modulus 1 eigenvalues. Besides unitary transformations, Zhang et al. (2018) controls the singular values of the transformation using SVD to avoid vanishing and exploding gradient problems. But, these approaches often impose computational

overhead to ensure the constraint and require non-trivial parameterizations. These approaches only address hidden-to-hidden gradients. In this work we discuss other properties, such as exploding hidden states in forward propagation when the input is constant for long periods.

We propose a sequence model called GATO. The key insight in GATO is that only part of the hidden state must have non-vanishing gradients for states in the far past to influence the current state. GATO splits its hidden state into two channels. At each time step, each channel is updated by a function of the input and current hidden state. To ensure that one of the channels will not have vanishing gradients, GATO limits the functions for both channels to only depend on one of the channels. This results in an identity-block structure in the hidden-hidden Jacobian. GATO then uses non-saturating functions, such as the cosine, to bound the hidden state upon input to the decoder to stabilize gradients for model parameters. We use a simple version of GATO where the dimensions within the channels do not interact and show that it can model arbitrary conditional distributions through memorization. Partitioned hidden states have been mentioned in (Oliva et al., 2017) and (Zhang et al., 2018). Oliva et al. (2017) can guarantee the identity Jacobian when $\alpha = 1$, but in this case the state stays constant.

We study GATO copying and adding on long sequences. GATO solves the copying task while the others do not (Figure 1). On adding, the GRU experiences difficulty as sequence length increases while GATO does not (Figure 2). We also test GATO on health and language data. Using a critical care database collected in Boston (Johnson et al., 2016), GATO predicts patient length-of-stay in the intensive care unit more accurately than GRUs and LSTMs across seeds and learning rates. GATO achieves lower held-out perplexity on language modeling using the Penn Treebank dataset. We explore this in a setting with no methods such as dropout or gradient clipping. GATO is stable to train across random initializations compared to GRUs, LSTMs, and other recently proposed RNNs such as the SRU (Oliva et al., 2017), EURNN (Jing et al., 2017), and RHN (Zilly et al., 2017).

## 2 BACKGROUND ON SEQUENCE MODELS

Sequence modeling targets learning the sequence of conditional distributions that constitute the joint. Let $\mathbf{x} = (\mathbf{x}_1, \mathbf{x}_2, \ldots, \mathbf{x}_T)$ be a sequence of $T$ observations with joint distribution $Q(\mathbf{x})$. We consider the standard probabilistic model $p_\theta(\mathbf{x})$ with parameter $\theta$, where each sequence element depends on all previous elements. Let $\mathcal{L}_t$ at timestep $t$ be the negative conditional log probability of the next sequence element according to the model:

$$p_\theta(\mathbf{x}) = p_\theta(\mathbf{x}_1) \prod_{t=1}^{T-1} p_\theta(\mathbf{x}_{t+1}|\mathbf{x}_{\leq t}); \qquad \mathcal{L}_t(\theta) = -\log p_\theta(\mathbf{x}_{t+1}|\mathbf{x}_{\leq t}); \qquad \mathcal{L}(\theta) = \sum_t \mathcal{L}_t(\theta).$$

To minimize $\mathcal{L}(\theta)$ on samples from $Q$, we need expressive $p_\theta$ to capture complicated conditionals.

### 2.1 SEQUENCE MODELING WITH NEURAL NETWORKS

Sequence models often represent conditionals using a hidden state that evolves as a sequence of data is observed. Using a function $D$ to map from states to distribution parameters, we define the conditional probability $p(\mathbf{x}_{t+1}|\mathbf{x}_{\leq t})$ with parameters $D(\mathbf{h}_t)$. To represent the true conditionals of $Q$, $\mathbf{h}_t$ must summarize the information in $\mathbf{x}_1 \ldots \mathbf{x}_t$ useful for predicting $\mathbf{x}_{>t}$. Typically, $D$ is chosen to be a feed-forward neural network. Let $\mathbf{U}$ be hidden-to-hidden parameters, $\mathbf{W}$ be data-to-hidden parameters, and $\sigma$ be an element-wise non-linearity. RNNs proceed by parameterizing the dynamics of the hidden state via the following discrete-time recurrence:

$$\mathbf{h}_{t+1} = \sigma(\mathbf{U}\mathbf{h}_t + \mathbf{W}\mathbf{x}_{t+1}). \tag{1}$$

Though RNNs can represent a broad class of sequences (Schäfer & Zimmermann, 2006; Siegelmann & Sontag, 1995), they face problems in modeling long sequences where an early part of the sequence influences a later part. On one hand, modeling such data requires the use of powerful functions, but on the other, gradient-based optimization of such functions is unstable and requires manual tuning.

### 2.2 RNNs HAVE PROBLEMS

**Vanishing and Exploding Gradients.** Consider two states $\mathbf{h}_t$ and $\mathbf{h}_T$ with times $t \ll T$. Exploding gradients occur when $h_T$ changes dramatically for small changes in $h_t$, while vanishing gradients occur when the influence of $h_t$ on $h_T$ decreases exponentially with gap $T - t$.

Consider the RNN as in Equation (1) and recall that $\sigma$ is an element-wise non-linearity. Let $\mathcal{L}_T$ be the loss at time $T$. Let $\mathbf{D}_t = \texttt{diag}(\sigma'(\mathbf{U}\mathbf{h}_{t-1} + \mathbf{W}\mathbf{x}_t))$ be the Jacobian for $\sigma$. Suppose that $\lambda$ is a

lower-bound on the absolute-value of the Jacobian $\mathbf{D}$'s entries. Then if the smallest singular value of $\mathbf{U}$ is larger than $1/\lambda$, $\partial \mathbf{h}_T/\partial \mathbf{h}_t$ blows up exponentially as time gap $T - t$ increases. Suppose instead that $\lambda$ is an upper-bound on the absolute-value of $\mathbf{D}$'s entries. Then if the largest singular value of $\mathbf{U}$ is smaller than $1/\lambda$, $\partial \mathbf{h}_T/\partial \mathbf{h}_t$ decays exponentially with time gap $T - t$. The Jacobian $\partial \mathbf{h}_T/\partial \mathbf{h}_t$ involved in $\mathcal{L}_T/\partial \mathbf{U}$ is

$$\frac{\partial \mathcal{L}_T}{\partial \mathbf{U}} = \sum_{t=1}^{T} \frac{\partial \mathcal{L}_T}{\partial \mathbf{h}_T} \frac{\partial \mathbf{h}_T}{\partial \mathbf{h}_t} \frac{\partial \mathbf{h}_t}{\partial \mathbf{U}},$$

and can stop losses $\mathcal{L}_T$ from backpropagating to $\mathbf{U}$. These issues cause short-term dependencies to be favored over long-term ones (Bengio et al., 1994; Chung et al., 2014).

**Exploding Forward Computation.** The hidden state can also grow without bound or vanish for long sequences. Consider the following recurrence where $\sigma$ denotes sigmoid:

$$\mathbf{z}^{(1)} = \sigma(\mathbf{W}^{(1)}\mathbf{x}_{t+1} + \mathbf{U}^{(1)}\mathbf{h}_t); \quad \mathbf{z}^{(2)} = \sigma(\mathbf{W}^{(2)}\mathbf{x}_{t+1} + \mathbf{U}^{(2)}\mathbf{h}_t);$$

$$\tilde{\mathbf{h}} = \texttt{tanh}(\mathbf{W}^{(3)}\mathbf{x}_{t+1} + \mathbf{U}^{(3)}\mathbf{h}_t); \quad \mathbf{h}_{t+1} = \mathbf{z}^{(1)} \odot \mathbf{h}_t + \mathbf{z}^{(2)} \odot \tilde{\mathbf{h}}.$$

The state $\mathbf{h}$ can grow without bound. Two examples of systems with good hidden-hidden gradients but that have unbounded forward propagation are RHN (Zilly et al., 2017) and (Efficient) Unitary RNN (Arjovsky et al., 2016; Jing et al., 2017). They are described in Appendix G and Appendix H. These models NAN during forward propagation in our experiments.

### 2.3 LONG SHORT-TERM MEMORY AND GATED RECURRENT UNITS

To improve gradients, Long Short-Term Memory (LSTM) (Hochreiter & Schmidhuber, 1997) and Gated Recurrent Units (GRU) (Cho et al., 2014) augment RNNs with additional computation that helps bound the state as well as reduce gradient vanishing. The LSTM retains a cell state that copies the hidden state forward in time (Appendix E). The GRU uses a convex combination of the current state and a bounded proposal. Its state is bounded when initialized correctly (Appendix F).

This variety of neural sequence models has produced state-of-the-art performance in machine translation (Sutskever et al., 2014; Luong et al., 2015) and image captioning (Xu et al., 2015). However, these models are difficult to optimize. Practitioners must experiment with optimization techinques including gradient clipping, randomly dropping-out hidden state updates (Krueger et al., 2016), and even training discriminators to keep generated and data sequences close (Lamb et al., 2016). Empirical investigation into the behavior of the LSTM gates has also shown that forget and input gate values tend toward $0.5$, contradicting explicit interpretations as gates (Li et al., 2018b). Recent analysis has shown that LSTMs still have gradient issues that cause them to favor learning short-term dependencies in data (Kanuparthi et al., 2019). See Appendix J for a brief discussion.

## 3 DESIDERATA FOR SEQUENCE MODELS

We now develop criteria for building sequence models and consider several important gradients.

**Hidden-to-Hidden gradients.** For sequence models to learn long-term dependencies, part of the hidden state at time $t$ needs to have non-decaying influence on states at later times $T$. Therefore only some, not all, of the hidden units need to avoid vanishing gradients. Our first criterion is:

    1. Part (a block) of the Jacobian $\partial \mathbf{h}_T/\partial \mathbf{h}_t$ does not depend on time gap $T - t$.

**Decoder Gradients.** Let $\gamma$ be a non-linear function and consider a loss at time $t$ defined through a linear multi-output decoder $D(\mathbf{h}_t) = \mathbf{W}\gamma(\mathbf{h}_t)$. Then the gradient of the loss with respect to $\mathbf{W}$ is:

$$\partial \mathcal{L}_t/\partial \mathbf{W} = (\partial \mathcal{L}_t/\partial D)(\partial D/\partial \mathbf{W}) = (\partial \mathcal{L}_t/\partial D)\gamma(\mathbf{h}_t)^{\top}$$

To ensure that $||\partial \mathcal{L}_t/\partial \mathbf{W}||$ does not become too large, one property that $\gamma$ needs to have is:

    2. $\gamma$ is bounded to ensure that decoder gradients $\partial D/\partial \mathbf{W}$ do not grow with time.

**Loss-to-Hidden Gradients.** $\partial \mathcal{L}_t/\partial \mathbf{h}_t$ for one time step $t$ also depends on the choice of $\gamma$ through its gradient $\gamma'$. It is defined through the decoder by

$$\partial \mathcal{L}_t/\partial \mathbf{h}_t = (\partial \mathcal{L}_t/\partial D)(\partial D/\partial \mathbf{h}_t) = \left( \mathbf{W}^{\top}(\partial \mathcal{L}_t/\partial D) \right) \odot \gamma'(\mathbf{h}_t)$$

This means that:

3. $\gamma'(\mathbf{h}_t)$ should not converge to $\mathbf{0}$ as $||\mathbf{h}_t||$ grows (if $\mathbf{h}_t$ does grow).

4. The magnitude $||\gamma'||$ evaluated at $\mathbf{h}_t$ is bounded even when $||\mathbf{h}_t||$ increases.

These criteria require bounded, differentiable functions whose gradients do not go to $\mathbf{0}$ as their input increases. There is a relationship between the particular form of $\mathbf{h}$'s update and the choice of $\gamma$. For example, if the update contracts to a constant $\mathbf{c}$ from both the positive and negative direction, then $\gamma$ should be well-behaved at $\mathbf{c}$ (e.g. non-zero gradients). These criteria rule out traditional choices of non-linear functions that saturate, such as `sigmoid` and `tanh`, and functions that grow such as `SoftPlus` and `ReLU`. Periodic functions such as `cos` and `sin` meet these criteria.

**Forward Computation.** In tasks such as language modeling, the hidden state can be processed for thousands of steps through the corpus. Let $\mathbf{r}_t$ denote the part of $\mathbf{h}_t$ involved in the recurrence to compute $\mathbf{h}_{t+1}$. Then if $||\mathbf{r}_t||$ is large at time $t$ during training, $||\partial \mathcal{L}_{t+1}/\partial \mathbf{U}||$ will be large and can cause additional issues in learning recurrent parameters $\mathbf{U}$.

5. $||\mathbf{r}_t||$ in general should not become large as $t$ increases.

## 4 GATO

We propose a new sequence model that meets the desiderata called GATO. GATO is capable of learning long-range dependencies and is stable to train, exhibiting little variance across learning rates and initializations. GATO requires fewer parameters to perform well on tasks such as copying, arithimetic, classification, and language modeling. We describe the model and analyze its gradients.

### 4.1 GATO MODEL

Building on the ideas in Section 3, GATO partitions the RNN hidden state into two parts, one of which is guaranteed to have good long-term gradients by construction. The updates of the hidden state impose a structure on the hidden-to-hidden Jacobian wherein a sub-matrix is the identity $\mathbf{I}$. This means that parts of $\partial \mathbf{h}_T/\partial \mathbf{h}_t$ do not dependent on $T - t$ and therefore do not vanish with time.

**Partitioned Hidden State.** The hidden state $\mathbf{h}_t$ has two channels $\mathbf{r}_t$ and $\mathbf{s}_t$. At each time step, both channels are updated by a non-linear function of the previous state. To ensure that one part of the state, $\mathbf{s}_t$, has time-independent hidden-to-hidden gradients, the updates do not depend on $\mathbf{s}$. Let $f_\phi$ and $g_\psi$ be non-linear functions with parameters $\phi$ and $\psi$. GATO's hidden state evolves as:

$$\mathbf{r}_{t+1} = f_\phi(\mathbf{r}_t, \mathbf{x}_{t+1}); \qquad \mathbf{s}_{t+1} = \mathbf{s}_t + g_\psi(\mathbf{r}_t, \mathbf{x}_{t+1}); \qquad \mathbf{h}_{t+1} = [\mathbf{r}_{t+1}, \mathbf{s}_{t+1}].$$

This is like an RNN where the hidden state update is only a function of part of the state, and where the updates of the other part are defined by residual connections (He et al., 2016; Wang & Tian, 2016). In our experiments, $\mathbf{r}$ and $\mathbf{s}$ are the same size. In Appendix A we experiment with $\mathbf{s}$ smaller than $\mathbf{r}$.

### 4.2 GATO HAS GOOD GRADIENTS

**Hidden-to-Hidden Gradients.** As shown in Section 2.2, RNNs suffer from vanishing gradients. We show that the GATO hidden state recurrence does not have vanishing gradients for part of the hidden state. First consider $\partial \mathbf{s}_T/\partial \mathbf{s}_t$ and recall the evolution of $\mathbf{r}$ and $\mathbf{s}$:

$$\mathbf{r}_T = f_\phi(f_\phi(...f_\phi(\mathbf{r}_t, \mathbf{x}_{t+1}))); \quad \mathbf{s}_T = \mathbf{s}_t + \sum_{k=t}^{T-1} g_\psi(\mathbf{r}_k, \mathbf{x}_{k+1}).$$

The right hand side for $\mathbf{r}_T$ and the second term of the right hand side for $\mathbf{s}_T$ do not depend on $\mathbf{s}_t$, so $\partial \mathbf{s}_T/\partial \mathbf{s}_t$ is the identity matrix $\mathbf{I}$. Next, $\partial \mathbf{r}_T/\partial \mathbf{s}_t = \mathbf{0}$ because $\mathbf{s}$ is not included among the arguments to the update $f_\phi(\mathbf{r}_t, \mathbf{x}_t)$. The whole Jacobian matrix of hidden states $\mathbf{h}_T$ with respect to $\mathbf{h}_t$ is:

$$\frac{\partial \mathbf{h}_T}{\partial \mathbf{h}_t} = \begin{bmatrix} \frac{\partial \mathbf{s}_T}{\partial \mathbf{s}_t} & \frac{\partial \mathbf{s}_T}{\partial \mathbf{r}_t} \\ \frac{\partial \mathbf{r}_T}{\partial \mathbf{s}_t} & \frac{\partial \mathbf{r}_T}{\partial \mathbf{r}_t} \end{bmatrix} = \begin{bmatrix} \mathbf{I} & \sum_{k=t}^{T-1} \frac{\partial g_\psi(\mathbf{r}_k, \mathbf{x}_{k+1})}{\partial \mathbf{r}_k} \frac{\partial \mathbf{r}_k}{\partial \mathbf{r}_t} \\ \mathbf{0} & \prod_{\ell=t}^{T-1} \frac{\partial f_\phi(\mathbf{r}_\ell, \mathbf{x}_{\ell+1})}{\partial \mathbf{r}_\ell} \end{bmatrix}. \tag{2}$$

Since $\partial \mathbf{s}_T/\partial \mathbf{s}_t = \mathbf{I}$ does not depend on time difference $T - t$, part of the hidden state, $\mathbf{s}_T$, retains its influence from $\mathbf{s}_t$ (criterion 1). Though $\partial \mathbf{s}_T/\partial \mathbf{r}_t$ and $\partial \mathbf{r}_T/\partial \mathbf{r}_t$ are complicated, $\partial \mathbf{s}_T/\partial \mathbf{s}_t$ is guaranteed not to vanish so gradients can be propagated back through time. In Appendix A, we explore the effect of removing this feature and find that model performance suffers.

**Decoder and Loss-to-Hidden Gradients.** We have just shown that the hidden state channel $\mathbf{s}_t$'s gradient does not vanish. As discussed in Section 3, there are additional optimization issues that can be introduced by the hidden state. We focus on $\mathbf{s}_t$. In the case of loss defined through decoder $D(\mathbf{s}_t) = \mathbf{W}\gamma(\mathbf{s}_t)$, $\gamma$ must be chosen to ensure that $||\partial\mathcal{L}_t/\partial\mathbf{W}||$ and $||\partial\mathcal{L}_T/\partial\mathbf{s}_t||$ do not shrink or grow, regardless of $||\mathbf{s}_t||$. In the experiments, GATO ensures this by using $\gamma(\mathbf{s}_t) = \cos(\mathbf{s}_t)$. We experiment with sin in Appendix B.2.

### 4.3 NON-INTERACTING HIDDEN UNITS

Consider a version of GATO where interactions are limited to pairs of units. Let $j$ index pairs of hidden units and let the hidden state size be $2J$. Then the updates for each pair $(r^{(j)}, s^{(j)})$ are:

$$r_{t+1}^{(j)} = f_\phi^{(j)}(r_t^{(j)}, \mathbf{x}_{t+1}); \qquad s_{t+1}^{(j)} = s_t^{(j)} + g_\psi^{(j)}(r_t^{(j)}, \mathbf{x}_{t+1}). \tag{3}$$

Non-interacting units in RNN hidden states have been explored previously (Li et al., 2018a). We show that this version of GATO can represent arbitrary conditionals by memorization. Consider a sequence model of scalar data with conditionals defined via hidden states $p(x_{t+1}|x_{\leq t}) = p(x_{t+1}; D(\mathbf{s}_t, \mathbf{r}_t))$. For this model to represent arbitrary conditionals, the hidden state needs to store $\mathbf{x}_{\leq t}$. One way for GATO in Equation (3) to memorize its inputs $\mathbf{x}$ would be for each $r^{(j)}$ to learn how to count and each function $g_\phi^{(j)}$ to store its one-dimensional input $x$ in $s^{(j)}$ when the count $r^{(j)}$ equals $j$. A powerful decoder can then read the memorized sequence from the hidden state and compute a conditional distribution. This construction requires the size $J$ to be at least the length of the sequence $T$.

### 4.4 TWO EXAMPLES OF NON-INTERACTING GATO

Let $\mathbf{x}_t \in \mathbb{R}^D$ denote the input at time $t$. Let $\mathbf{h}_t = [\mathbf{r}_t, \mathbf{s}_t]$ denote the hidden state with $\mathbf{r}_t, \mathbf{s}_t \in \mathbb{R}^J$. Let $\mathbf{A}, \mathbf{B}, \mathbf{C} \in \mathbb{R}^{J \times D}$ and $\mathbf{a}, \mathbf{b}, \mathbf{c} \in \mathbb{R}^J$ be parameters. Let $\sigma$ be a nonlinear function whose maximum absolute value is 1. Let $\lambda \in \mathbb{R}$.

**One layer GATO.**

$$\begin{aligned}
\mathbf{s}_{t+1} &= \mathbf{s}_t + \texttt{SoftPlus}(\mathbf{A}\mathbf{x}_{t+1} + \mathbf{a} \odot \mathbf{r}_t); \\
\mathbf{r}_{t+1} &= \lambda \cdot \sigma(\mathbf{B}\mathbf{x}_{t+1} + \mathbf{b} \odot \mathbf{r}_t) \odot \mathbf{r}_t + \texttt{tanh}(\mathbf{C}\mathbf{x}_{t+1} + \mathbf{c} \odot \mathbf{r}_t).
\end{aligned} \tag{4}$$

**Two layer GATO.** Let $\texttt{F}(\mathbf{r}, \mathbf{x})$ denote a neural network with inputs $\mathbf{r}, \mathbf{x}$ such that each output dimension $F_j$ is a one hidden layer neural network with ReLU activation and hidden size $k$ that only depends on $r_j$ and $\mathbf{x}$. Then two-layer GATO is defined as:

$$\begin{aligned}
\mathbf{s}_{t+1} &= \mathbf{s}_t + \texttt{SoftPlus}(\texttt{F}(\mathbf{x}_{t+1}, \mathbf{r}_t)); \\
\mathbf{r}_{t+1} &= \lambda \cdot \sigma(\mathbf{A}\mathbf{x}_{t+1} + \mathbf{a} \odot \mathbf{r}_t) \odot \mathbf{r}_t + \texttt{tanh}(\mathbf{B}\mathbf{x}_{t+1} + \mathbf{b} \odot \mathbf{r}_t).
\end{aligned} \tag{5}$$

We call the $\lambda \cdot \sigma(.)$ term the regularizer. In both models, setting $\lambda$ to be a number less than one helps meet criterion 5 in Section 3. To see this, assume $\lambda = 0.7$ and that $\sigma$ takes its maximum value 1:

$$\mathbf{r}_{t+1} = 0.7 \cdot \mathbf{r}_t + \texttt{tanh}(\mathbf{B}\mathbf{x}_{t+1} + \mathbf{b} \odot \mathbf{r}_t).$$

Now assume that the tanh takes on either of its extremes, for example 1. Then the update is $r = 0.7 \cdot r + 1$ and any dimension of $\mathbf{r}$ that is greater than $10/3$ will be shrunk towards $10/3$. We set $\sigma$ to the sigmoid in the main experiments (Section 5) and to tanh in Appendix B.1.

## 5 EXPERIMENTS

We study GATO on two standard sequence tasks, copying and performing arithmetic on long sequences (Hochreiter & Schmidhuber, 1997; Arjovsky et al., 2016). We compare against several kinds of RNNs, ranging from well-established to recently proposed alternatives. In these tasks, the models train directly on new samples from the data distribution at each batch. There is no notion of overfitting on these experiments. These experiments show where common sequence models fail.

We then use GATO for hospital length of stay prediction using electronic health records and on language modeling. For these tasks, we report held-out metrics (accuracy and perplexity), though we do not focus on regularizing sequence models in this work. We explore this in a setting with no gradient clipping (which could stabilize training) or dropout (which could reduce overfitting). We run experiments across three random seeds and choices of learning rates.

## 5.1 MODEL AND BASELINE SPECIFICATIONS

For baselines, we use PyTorch modules `nn.GRU` and `nn.LSTM` (Paszke et al., 2017). Since GATO has fewer parameters than these models for a given hidden size, we consider two alternative baselines. We use diagonal weight matrices for LSTM and GRU (GATO's weight matrices are also diagonal). We call these models GRU-DIAG and LSTM-DIAG. When we set them to have the same number of parameters as GATO rather than the same hidden size, we call them GRU-DIAG (same param) and LSTM-DIAG (same param). We also compare against SRU (Appendix I), RHN (Appendix G), and EURNN (Appendix H). For GATO, we choose $\sigma$ in the $r$ update regularization term to be the `sigmoid` and choose $\lambda = 0.7$. In our experiments, we find $\lambda = 0.5$ performed similar to 0.7.

We give exact numbers of RNN parameters for each experiment. Embedding and decoder parameters are constant across models and excluded in counts. More details including initialization are described in Appendix C. All models are trained across three random seeds for all tasks. The error bands in all plots represent the minimum and maximum across seeds with a bold line representing the mean.

## 5.2 EXPERIMENT 1: LONG COPY TASK

**Task Description.** We model sequences of the form `ABA`, where `A` is a sequence of length 20 with tokens uniformly drawn from $\{1, \ldots, 10\}$ and `B` is 100 constant `BLANK` tokens. The model must predict the second `A` as a copy of the first. The hidden state must memorize the first 20 tokens and keep them in memory while processing the 100 `BLANK`'s. This is a harder variant of the copy task in (Arjovsky et al., 2016). We do not use a special token to mark the beginning of the second A region.

**Experiment Setup.** All models use an input embedding size of 4, recurrent hidden size of 1024, and a fully-connected decoder with one hidden layer of size 256. We use two layer GATO. GATO has $138,752$ parameters, while the GRU and LSTM have $3,118,080$ and $4,222,976$, respectively. We draw new sequences from the data distribution for each step of training. We evaluate on a held-out set of 1000 sequences. We let each model see 1 million training points.

**Results.** We show the results in Figure 1. For the LSTM and GRU, the probability of the tokens to be copied is near chance (0.10) across the three seeds. GATO learns the task with at least .50 probability averaged across the held-out set for all tokens to be copied. GRU-DIAG beats other baselines when excluding EURNN. EURNN NAN'ed on one seed (not shown) and was unstable for the other two.

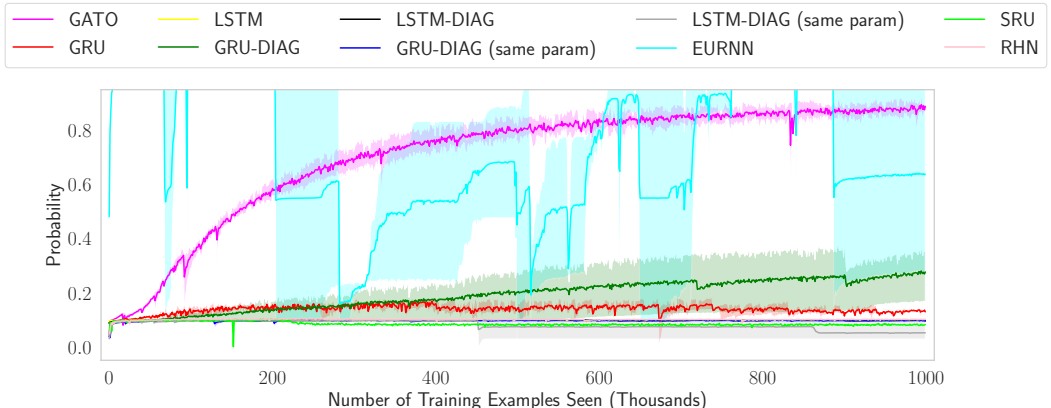

Figure 1: **GATO learns to copy long sequences with an order of magnitude fewer parameters than the GRU and LSTM.** We plot average probability of copied tokens on a held-out set. Error bands correspond to random seeds. The GRU and LSTM output probabilities near chance for the copied tokens. GRU-DIAG with same number of parameters as GATO beats other baselines when excluding EURNN. EURNN NANed on one seed (not shown) and was unstable for the other two.

## 5.3 EXPERIMENT 2: ADDING PROBLEM

**Task Description.** This experiment follows the adding task described in (Hochreiter & Schmidhuber, 1997). In this task, the input consists of two sequences of length $T$. Each element of the first sequence is sampled from the uniform distribution $\mathcal{U}[0, 1]$. The second sequence plays the role of an indicator. It is all 0's except for two 1's. The location of the first 1 is sampled uniformly from the first half of the sequence, while the location of the second 1 is sampled uniformly from the second half of

the sequence. The task is to predict the sum of the two numbers in the first sequence whose locations correspond to the locations of the 1's in the second sequence. The mean squared error of predicting the expectation, 1, is 0.167. This forms the baseline.

**Experiment Setup.** We compare all models with the baseline and we use two-layer GATO. Loss is measured by mean squared error. We do not use embeddings for this task, so the input size at each timestep is 2 (uniform variate and indicator). All models use recurrent hidden size 512 and a decoder with one hidden layer of size 256. For recurrent parameters, GATO has 43,264 parameters while the GRU and LSTM have 792,576 and 1,056,768 respectively. To make a prediction, we apply the decoder to the RNN hidden state after processing the sequence. We sample new data points at each step of training. We evaluate on a held-out set of 1000 examples. We run experiment with sequence lengths $T = 100, 200, 400, 750$.

**Results.** Our results are shown in Figure 2. After training with 200,000 examples, the mean squared error of the LSTM and both LSTM-DIAGs was still around the baseline. GRU and the GRU-DIAGs learned for T=100 and 200. On T=400, GRU had a NAN error on one seed (denoted by infinite loss). The GRU did not learn the task on T=750, but GRU-DIAG did. Neither SRU nor RHN could learn the task, with missingness in the plot resulting from infinite and NAN losses. GATO (magenta line) learned this task at all lengths and was stable across random seeds, also learning with fewer examples.

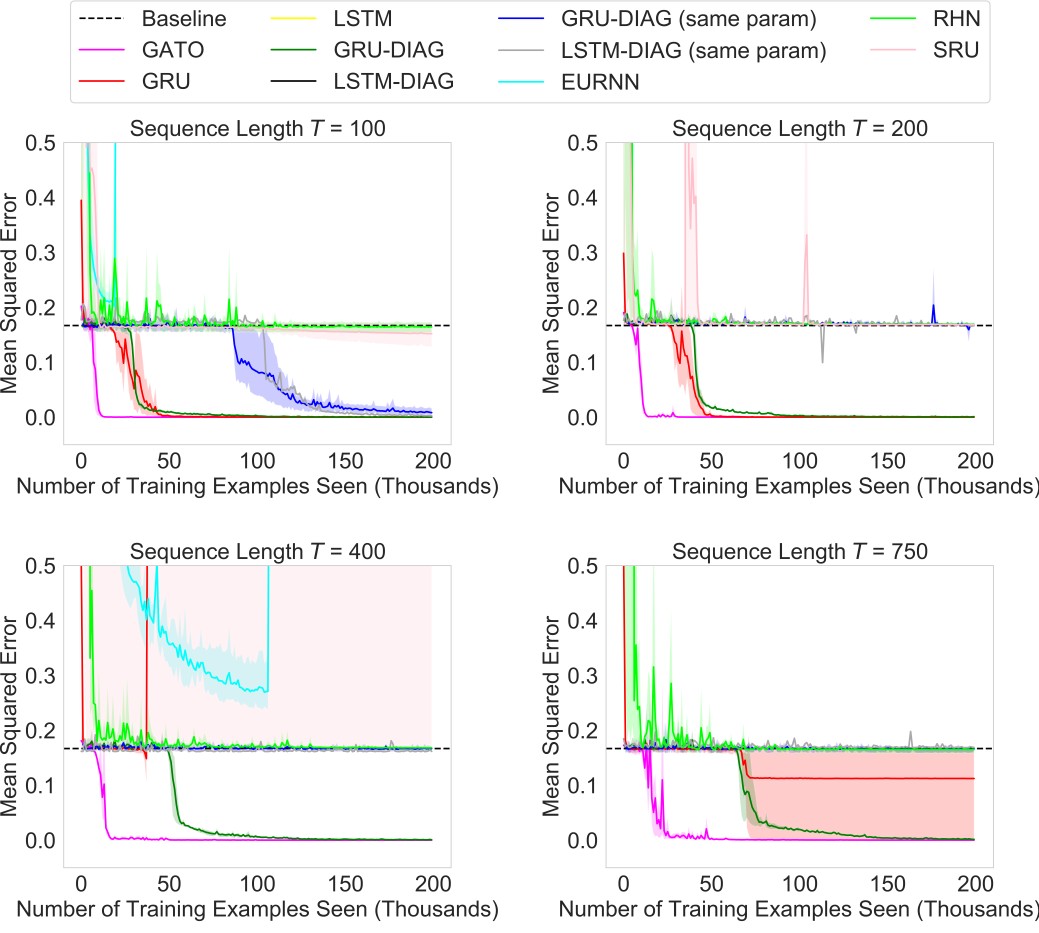

Figure 2: **GATO is more robust across different seeds and sequence lengths.** We show the mean squared errors of the adding problem for sequence lengths $T = 100, 200, 400, 750$ over number of training examples. LSTM failed to learn the task. For length $T = 400$, the GRU numerically failed on one seed (NAN error). We denote this by infinite error. GRU-DIAG could learn the task with more samples than GATO. SRU, EURNN, and RHN could not learn this task (NAN and/or worse than chance). NANs and errors greater than 0.5 (significantly worse than chance) are not shown (e.g. EURNN, T=750).

## 5.4 Experiment 3: Length of Stay in the Intensive Care Unit (ICU)

**Task Description and Experimental Setup.** This experiment uses MIMIC-III, a publicly-available Boston-based critical care database. The task is to predict if a patient stays more than 1 week in the ICU unit using the first 48 hours of their vitals. There are 53,211 patients in total, 27,088 of whom stay more than one week. We use the vitals extracted for Interpolation-Prediction Networks (Shukla & Marlin, 2019). See Appendix C for more details. We split the data into a training set of size 43,000, validation set of size 5,106, and test set of 5,105. We use cross entropy loss. To make a prediction, we apply a decoder to the RNN hidden state after processing the sequence of vital signs. We use a recurrent hidden size 512 and decoder hidden size 1024 for all models. We use one-layer GATO.

**Results.** We report average test accuracy across seeds in Table 1. GATO has higher accuracy than GRU, LSTM, and their diagonal variants on this task across two learning rates. RHN performs in between the GRU and LSTM variants. SRU performs best on this task, but has more parameters for the same hidden size. The ranges in performance for GATO and SRU overlap.

Table 1: **GATO performs well on test accuracy with both learning rates 4e-3 and 1e-4.** We show the accuracy of length of stay prediction averaged across seeds. SRU performs best on this task. The ranges in performance for GATO and SRU overlap. *EURNN had an NAN error on all seeds shortly following the reported accuracies.

| Model | # Parameters | Test Accuracy, LR 4e-3 | Test Accuracy, LR 1e-4 |
|---|---|---|---|
| GATO | 20,736 | 0.693 | 0.646 |
| GRU | 827,904 | 0.519 | 0.569 |
| LSTM | 1,103,872 | 0.650 | 0.569 |
| GRU-DIAG | 41,472 | 0.505 | 0.506 |
| LSTM-DIAG | 55,296 | 0.633 | 0.506 |
| GRU-DIAG (same params) | 21,504 | 0.570 | 0.505 |
| LSTM-DIAG (same params) | 22,400 | 0.569 | 0.506 |
| SRU | 315,282 | 0.700 | 0.683 |
| EURNN | 14,848 | 0.658* | 0.610* |
| RHN | 827,904 | 0.619 | 0.570 |

## 5.5 Experiment 4: Language Modeling on Penn TreeBank

**Description and Results.** A language model predicts the probability distribution of the next word in a sentence based on the previous words. We test GATO on the Penn Treebank dataset preprocessed by Mikolov et al. (2010). We choose hidden size 1300, embedding size 650 and a linear decoder for all the models for this task. We use one-layer GATO in this experiment. Table 2 shows model perplexity. GATO achieves similar perplexity to GRU. RHN gets infinite loss because of its unbounded hidden state and the long sequence lengths on this task. Excluding the infinite loss, SRU and EURNN have the highest test perplexity. By applying principles from GATO, we bound RHN's hidden state and call it BRHN. BRHN performs similarly to GATO and GRU.

## 6 Conclusion and Future Work

In this work, we study criteria for building sequence models that can capture long-term dependencies. We introduce one such model, GATO. GATO uses partitioned hidden states. One part of the hidden state is modified, but never used to update future states. This ensures that part of the hidden state does not have vanishing gradients. We explore a variant of GATO with non-interacting hidden units and show it is sufficient in our experiments. We also introduce criteria for avoiding unbounded forward propagation in sequence models. Empirically, such explosion occurs in our baseline models. GATO performs as well as GRUs and LSTMs in four typical sequence modeling problems: copying, arithmetic tasks with long-term dependencies, classification, and language modeling. We made similar comparisons against GRU and LSTM by matching the number of parameters instead of the hidden size. All alternative models (SRU,RHN,EURNN), demonstrate significant instabilities. In RHN, we identify that its hidden state is unbounded and use principles from this work to improve its performance.

Table 2: **GATO outperforms GRU and LSTM.** We calculate the perplexity averaged over three seeds on Penn Treebank language modeling in the unregularized setting. GATO achieves similar perplexity to GRU. RHN gets infinite loss. Excluding the RHN, SRU and EURNN have the highest test perplexity. *By applying principles from GATO, we bound RHN's hidden state to make it perform similarly to GATO and GRU (called BRHN).

| Model | # Parameters | Test Perplexity Score |
|---|---|---|
| GATO | 1,273,350 | 112.85 |
| GRU | 7,612,800 | 115.78 |
| LSTM | 10,150,400 | 122.86 |
| GRU-DIAG | 2,546,700 | 124.44 |
| LSTM-DIAG | 3,395,600 | 119.09 |
| GRU-DIAG (same params) | 1,273,350 | 124.65 |
| LSTM-DIAG (same params) | 1,306,000 | 122.01 |
| SRU | 2,198,560 | 149.97 |
| EURNN | 850,200 | 768.96 |
| RHN | 7,612,800 | $\infty$ |
| BRHN* | 5,075,200 | 116.11 |

Future directions include understanding when forward propagation is stable and how the principles used to develop GATO can be used to design feed-forward neural networks. Since several years of research have been devoted to investigating regularization of GRUs and LSTMs, an important direction is to see whether the same techniques apply to alternate models such as GATO, SRU, EURNN, and others. Studying combinations of GATO and other (deep) variants of RNNs such as Phase LSTM (Neil et al., 2016), Dilated RNN (Chang et al., 2019), Skip RNN (Campos et al., 2018), Fast-slow RNN (Mujika et al., 2017) and Deep RNN (Pascanu et al., 2013) is another interesting direction.

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

# A    GATO CANNOT LEARN ADDING WITHOUT LONG-TERM GRADIENT

Recall that GATO partitions its hidden state as $\mathbf{h}_t = [\mathbf{s}_t, \mathbf{r}_t]$ and that $\mathbf{s}_t$ and $\mathbf{r}_t$ are updated by:

$$\mathbf{r}_{t+1} = f_\phi(\mathbf{x}_{t+1}, \mathbf{r}_t); \qquad \mathbf{s}_{t+1} = \mathbf{s}_t + g_\psi(\mathbf{x}_{t+1}, \mathbf{r}_t).$$

We experiment with two ablations of GATO for the adding task and show that the long-term gradients through $\partial \mathbf{s}_T / \partial \mathbf{s}_t$ are necessary for its performance.

1. Set the update for $\mathbf{s}$ to $\mathbf{0}$. This makes GATO close to a standard RNN for half of its state.

2. Set the update for $\mathbf{s}$ to $g_\psi$. This is a standard RNN with half of its state but where the decoder can use $g_\psi$ to compute an additional function of the state $\mathbf{r}$.

Both of these ablations lose GATO's identity block matrix in the Jacobian $\partial \mathbf{h}_T / \partial \mathbf{h}_t$. In these experiments, GATO does not learn the adding task for sequence length 750 above chance for either ablation, across three seeds. We perform a third ablation by keeping $\mathbf{s}$, but reduce its size:

3. Set $\mathbf{s}$ to be $1/4$ of the hidden state instead of $1/2$.

We find that $1/4$ is less stable for the adding task (Figure 3) and performs worse for Penn TreeBank (Table 3). This suggests that having a larger fraction of the state devoted to long-term gradient propagation is important.

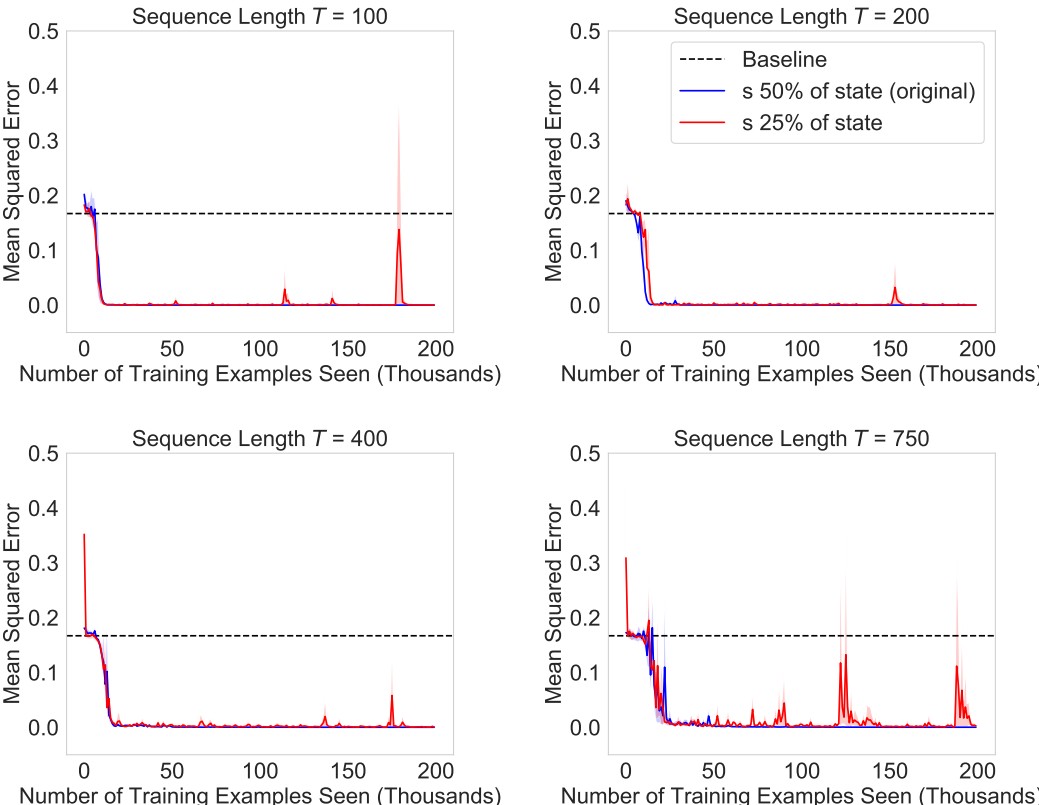

Figure 3: **Both sizes of s work well for the add task.** 50% is more stable, suggesting long-term gradient propagation is important for this task. This is consistent with the findings in Appendix A where 0% **s** failed on this task.

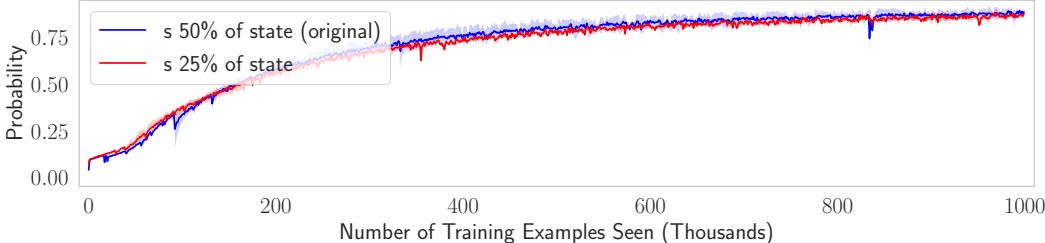

Figure 4: **Both sizes of s work well for the copy task.**

Table 3: **Size of s can matter**. For MIMIC, using $1/4$ of the hidden state for **s** works slightly better than using $1/2$ as in the main experiments. For Penn Treebank, $1/2$ was better than $1/4$, suggesting long-term gradient propagation is important for this task.

| Model | MIMIC Test Acc., LR 4e-3 | MIMIC Test Acc., LR 1e-4 | Penn Test PPL |
|---|---|---|---|
| 1/2 state (original) | **0.693** | **0.646** | **112.85** |
| 1/4 state | 0.696 | 0.651 | 122.293 |

# B ABLATION STUDY: GATO'S NON-LINEARITIES

## B.1 NON-LINEARITY IN UPDATE FOR R

Though $s$ is guaranteed to have good gradients, care must still be taken when choosing the update for $r$ as in any recurrent model. Recall the update for $r$:

$$\mathbf{r}_{t+1} = \lambda \cdot \sigma(\mathbf{B}\mathbf{x}_{t+1} + \mathbf{b} \odot \mathbf{r}_t) \odot \mathbf{r}_t + \tanh(\mathbf{C}\mathbf{x}_{t+1} + \mathbf{c} \odot \mathbf{r}_t). \tag{6}$$

We experiment by setting the $\sigma$ in the regularizing term to be `tanh` instead of sigmoid. Performance is similar to `sigmoid` on Copy (Figure 5), Add (Figure 6), MIMIC (Table 4), and Penn TreeBank (Table 4)

## B.2 DECODER NON-LINEARITY FOR S

The criteria in Section 3 suggest bounding the hidden state with periodic functions before decoding. Our experiments used `cos`. We believe that these criteria are necessary, but not necessarily sufficient. We also try `sin` and observe no change in performance on Copy (Figure 7), Add (Figure 8), MIMIC (Table 5), and Penn TreeBank (Table 5). It is unclear which other functions meet these criteria. A next step would explore finite sums of `cos` and `sin`.

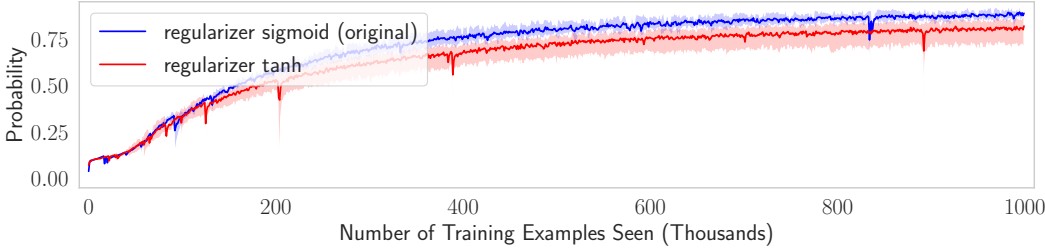

Figure 5: **Tanh performs similarly to sigmoid for regularizing non-linearity on copy.**

Table 4: **Tanh performs similarly to sigmoid for regularizing non-linearity on MIMIC and Penn TreeBank.**

| Model | MIMIC Test Acc., LR 4e-3 | MIMIC Test Acc., LR 1e-4 | Penn Test PPL |
|---|---|---|---|
| `sigmoid` (original) | **0.693** | **0.646** | **112.85** |
| `tanh` | 0.693 | 0.646 | 118.955 |

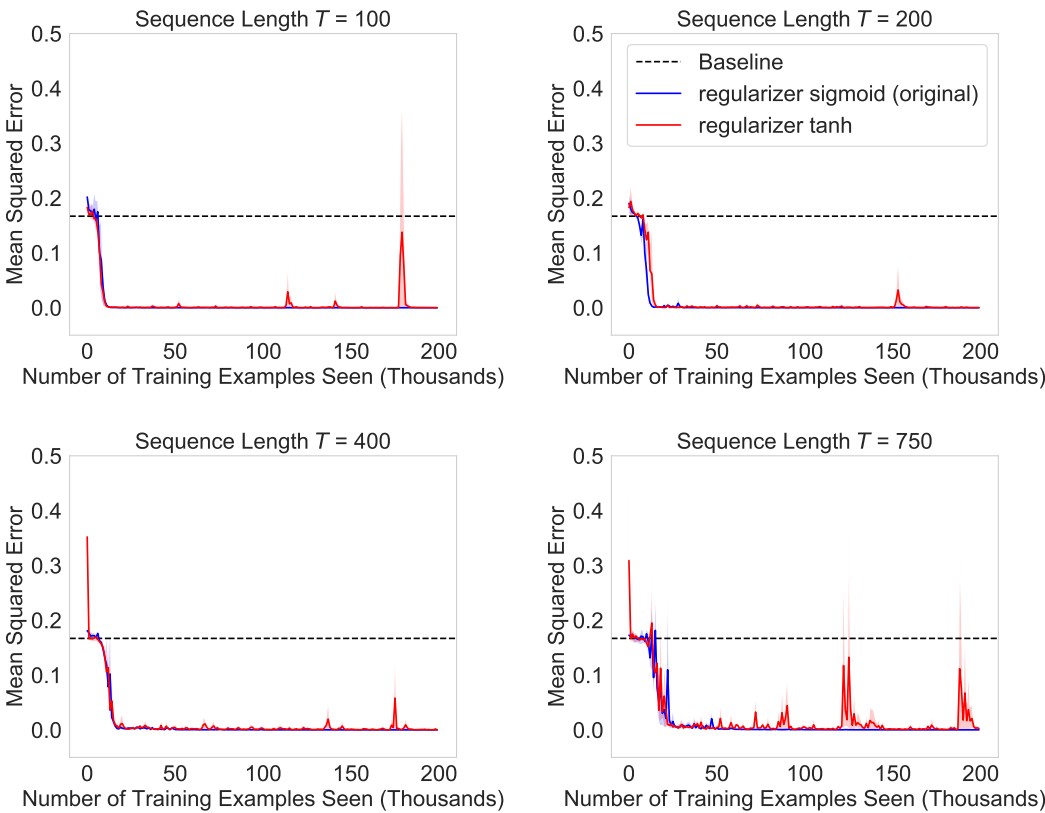

Figure 6: **Tanh performs similarly to sigmoid for regularizing non-linearity on add.**

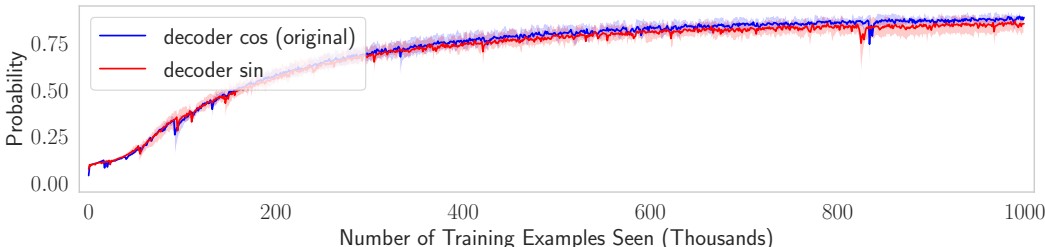

Figure 7: **sin for decoder non-linearity performs similarly to sos on copy task.**

Table 5: **sin for decoder non-linearity performs similarly to cos on MIMIC and Penn TreeBank**

| Model | MIMIC Test Acc., LR 4e-3 | MIMIC Test Acc., LR 1e-4 | Penn Test PPL |
|---|---|---|---|
| cos (original) | **0.693** | **0.646** | **112.85** |
| sin | 0.692 | 0.648 | 112.89 |

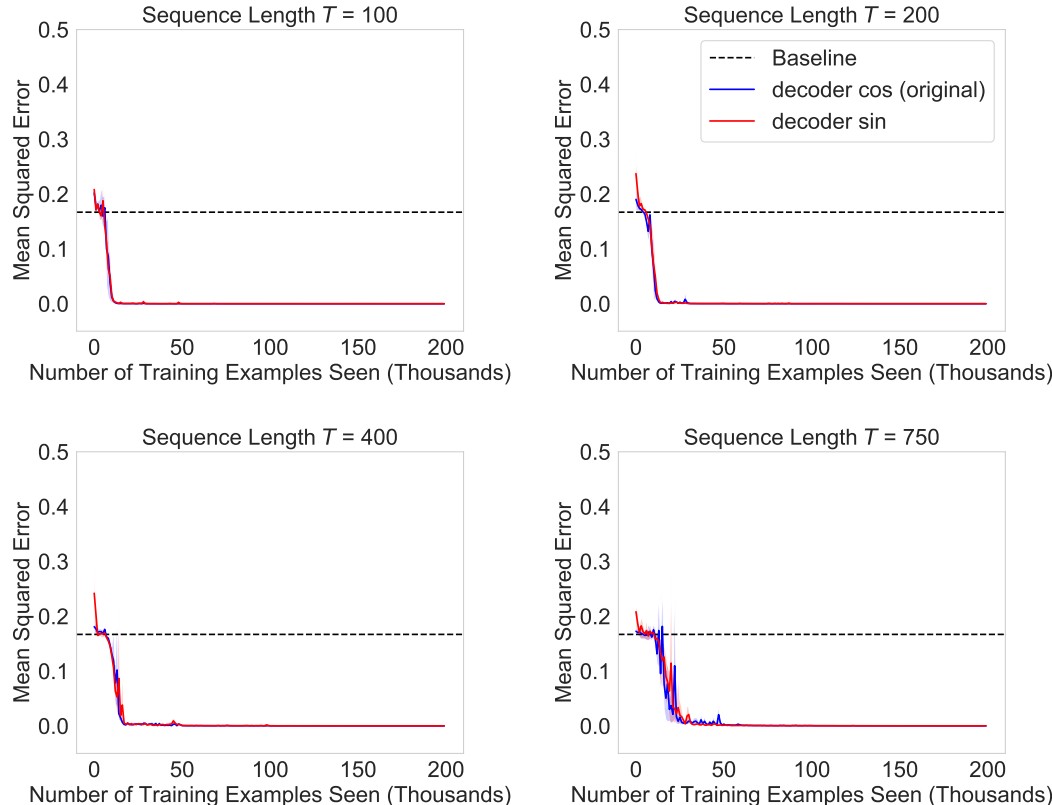

Figure 8: **sin for decoder non-linearity performs similarly to cos on adding task.**

## C EXPERIMENTAL DETAILS

### C.1 MIMIC-III

**Features.** To predict hospital length of stay on the MIMIC-III dataset, we use the same features extracted for Interpolation-Prediction Networks (Shukla & Marlin, 2019). The features include blood oxygen level, heart rate, respiratory rate, systolic blood pressure, diastolic blood pressure, temperature, oxygen saturation, and others. Not all features are recorded at each observation time. We use 25 features including 12 vitals, 12 additional missingness indicators for those features, and a measurement time.

**Continuous Embeddings.** All models apply a linear transformation followed by an element-wise `tanh` function to the continuous input before it is passed to the RNN. The linear transformation is learned alongside other model parameters.

### C.2 INITIALIZATION

Recall that experiment 1 is copying, experiment 2 is adding, experiment 3 is hospital stay prediction, and experiment 4 is language modeling on Penn Treebank. We use different initializations for GATO in experiments 1 and 2 than we do for experiments 3 and 4. In experiments 1 and 2, we use initialization $\mathcal{U}[-0.1, 0.1]$ for all RNN parameters in GATO. In experiments 3 and 4, we use initialization $\mathcal{U}[-0.1, 0.1]$ for all RNN parameters besides the weight matrices in the linear layers used to project $\mathbf{x}$. For those, we use the default PyTorch initializations. For embeddings and decoders for all models, as well as all GRU and LSTM parameters, we use default PyTorch initializations. We use initialization $\mathcal{U}[-0.1, 0.1]$ for parameters in RHN, PyTorch initializations for the linear layers in SRU and the linear layer for inputs in EURNN. For the representation of unitary matrices parameters in EURNN, we use standard normal initializations.

### C.3 LEARNING RATE AND OPTIMIZER

We use Adam optimizer for all experiments. We use learning rate 4e-3 for experiments 1 and 2, and 1e-4 for experiment 4. We check both learning rates 4e-3 and 1e-4 for experiment 3. In experiment 2,

we start with learning rate 0.004 and check the training loss after seeing every 10,000 examples. If the training loss is larger than for the previous 10,000 examples, we divide the learning rate by 2.

### C.4 VALIDATION AND TEST SCORES

On MIMIC-III and Penn Treebank, we report test set accuracy corresponding to each model's predictions when they achieved their best validation accuracy. Perplexity on the language modeling task is defined as in Merity et al. (2018).

### C.5 BATCH SIZE

We use batch size 32, 64, 256, and 20 in experiment 1, 2, 3, and 4 respectively.

### C.6 WEIGHT DECAY

We use 1.2e-6 weight decay (the default number in Merity et al. (2018)) in experiments 3 and 4. Since there is no generalization problem in experiments 1 and 2 (because data samples are drawn from the true generating distribution), we do not use weight decay in those experiments.

### C.7 LANGUAGE MODEL TRAINING

We follow the training strategy in (Merity et al., 2018). With probability 0.95, we take a batch with sentence length sampled from $\mathcal{N}(70, 5^2)$ (then rounded to an integer) and with probability 0.05 we take a batch with sentence length sampled from $\mathcal{N}(35, 5^2)$ (then rounded to an integer).

### C.8 OTHER TRICKS

We do not use dropout, gradient clipping and other tricks in all the experiments.

## D VANISHING AND EXPLODING GRADIENTS

Consider the RNN as in Equation (1) and recall that $\sigma$ is an element-wise non-linearity:

$$\mathbf{h}_{t+1} = \sigma(\mathbf{U}\mathbf{h}_t + \mathbf{W}\mathbf{x}_{t+1}).$$

Let $\mathcal{L}_T$ be the loss at time $T$. Let $\mathbf{D}_t = \texttt{diag}(\sigma'(\mathbf{U}\mathbf{h}_{t-1} + \mathbf{W}\mathbf{x}_t))$ be the Jacobian for $\sigma$.

Then the gradient of the loss $\mathcal{L}_T$ with respect to $\mathbf{U}$ is:

$$\frac{\partial \mathcal{L}_T}{\partial \mathbf{U}} = \sum_{t=1}^{T} \frac{\partial \mathcal{L}_T}{\partial \mathbf{h}_T} \frac{\partial \mathbf{h}_T}{\partial \mathbf{h}_t} \frac{\partial \mathbf{h}_t}{\partial \mathbf{U}} = \sum_{t=1}^{T} \left( \left( \prod_{k=t}^{T} \mathbf{U}^\top \mathbf{D}_k \right) \frac{\partial \mathcal{L}_T}{\partial \mathbf{h}_T} \mathbf{h}_{t-1}^\top \right).$$

Suppose that $\lambda$ is a lower-bound on the absolute-value of the Jacobian $\mathbf{D}_k$'s entries. Then if the smallest singular value of $\mathbf{U}$ is larger than $1/\lambda$, $\partial \mathbf{h}_T / \partial \mathbf{h}_t$ blows up exponentially with time gap $T - t$. This is called the exploding gradient problem. Suppose instead that $\lambda$ is an upper-bound on the absolute-value of $\mathbf{D}_k$'s entries. Then if the largest singular value of $\mathbf{U}$ is smaller than $1/\lambda$, $\partial \mathbf{h}_T / \partial \mathbf{h}_t$ decays exponentially with time gap $T - t$. This is called the vanishing gradient problem. The term $\partial \mathbf{h}_T / \partial \mathbf{h}_t$ in the gradient calculation for $\partial \mathcal{L}_T / \partial \mathbf{U}$ is therefore problematic for models like the RNN. Such issues can stop losses $\mathcal{L}_T$ at later time steps $T$ from backpropagating to $\mathbf{U}$ to adjust the computation of earlier states $\mathbf{h}_t$ for $t \ll T$.

## E LSTM

The LSTM (Hochreiter & Schmidhuber, 1997) maintains an auxlary cell state that helps copy the hidden state forward in time as a sequence is processed. $\mathbf{c}_t$ is called the cell state, and $\mathbf{h}_t$ is called the hidden state. Letting $\mathbf{f}_t, \mathbf{i}_t, \mathbf{g}_t$ and $\mathbf{o}_t$ be non-linear functions of $h_{t-1}$ and $\mathbf{x}_t$ and letting $\odot$ denote element-wise product, the LSTM hidden state computation for $\mathbf{c}$ and $\mathbf{h}$ is defined by

$$\mathbf{c}_t = \mathbf{f}_t \odot \mathbf{c}_{t-1} + \mathbf{i}_t \odot \mathbf{g}_t; \qquad \mathbf{h}_t = \mathbf{o}_t \odot \texttt{tanh}(\mathbf{c}_t). \tag{7}$$

In the update of the cell state, the *forget* gate $\mathbf{f}_t$ determines the influence of previous cell states on the next. The *input* gate $\mathbf{i}_t$ determines the influence of new information on the cell. $\mathbf{f}_t, \mathbf{i}_t, \mathbf{o}_t$ and $\mathbf{g}_t$ are each defined via usual RNN recurrences:

$$\begin{aligned}
\mathbf{g}_t &= \texttt{tanh}(\mathbf{W}_{gh}\mathbf{h}_{t-1} + \mathbf{W}_{gx}\mathbf{x}_t + \mathbf{b}_g), & \mathbf{f}_t &= \sigma(\mathbf{W}_{fh}\mathbf{h}_{t-1} + \mathbf{W}_{fx}\mathbf{x}_t + \mathbf{b}_f), \\
\mathbf{i}_t &= \sigma(\mathbf{W}_{ih}\mathbf{h}_{t-1} + \mathbf{W}_{ix}\mathbf{x}_t + \mathbf{b}_i), & \mathbf{o}_t &= \sigma(\mathbf{W}_{oh}\mathbf{h}_{t-1} + \mathbf{W}_{ox}\mathbf{x}_t + \mathbf{b}_o).
\end{aligned}$$

## F  GATED RECURRENT UNITS

The GRU (Cho et al., 2014) is similar to the LSTM but with fewer gates and parameters. First, two RNN-like functions $\mathbf{r}$ and $\mathbf{z}$ of inputs $\mathbf{x}_t$ and $\mathbf{h}_{t-1}$ are computed. They are called the reset and update gates, respectively. Then, a third RNN-like function with inputs $\mathbf{x}_t, \mathbf{h}_{t-1}$, and $\mathbf{r}$ computes a proposed state $\tilde{\mathbf{h}}$. Finally, the new state is a convex combination of the previous and proposed states. Let $\sigma$ be the `sigmoid` and $\phi$ be any non-linearity like the `tanh`. Then the GRU computation is:

$$\mathbf{r} = \sigma\left(\mathbf{W}_r\mathbf{x}_t + \mathbf{U}_r\mathbf{h}_{t-1}\right)$$
$$\mathbf{z} = \sigma\left(\mathbf{W}_z\mathbf{x}_t + \mathbf{U}_z\mathbf{h}_{t-1}\right)$$
$$\tilde{\mathbf{h}} = \phi\left(\left(\mathbf{W}\mathbf{x}_t + \mathbf{U}\left(\mathbf{r} \odot \mathbf{h}_{t-1}\right)\right)\right)$$
$$\mathbf{h}_t = \mathbf{z} \odot \mathbf{h}_{t-1} + (1 - \mathbf{z}) \odot \tilde{\mathbf{h}}$$

The expression for $\mathbf{h}_t$ is a convex combination of $\mathbf{h}_{t-1}$ and proposed update $\tilde{\mathbf{h}}$. Therefore, if $\phi$ is chosen to be bounded by 1 and if $\mathbf{h}$ starts with all coordinates in $(0, 1)$, the state will stay bounded.

## G  RECURRENT HIGHWAY NETWORK (RHN)

One-layer RHNs (Zilly et al., 2017) are another alternative gated recurrent networks. Define $\mathbf{h}_t, \mathbf{x}_t$ to be hidden states and inputs at time $t$. The update in the RHNs computation is:

$$\mathbf{r} = \sigma\left(\mathbf{W}_r\mathbf{x}_t + \mathbf{U}_r\mathbf{h}_{t-1}\right)$$
$$\mathbf{z} = \sigma\left(\mathbf{W}_z\mathbf{x}_t + \mathbf{U}_z\mathbf{h}_{t-1}\right)$$
$$\tilde{\mathbf{h}} = \texttt{tanh}\left(\mathbf{W}\mathbf{x}_t + \mathbf{U}\mathbf{h}_{t-1}\right)$$
$$\mathbf{h}_t = \mathbf{r} \odot \mathbf{h}_{t-1} + \mathbf{z} \odot \tilde{\mathbf{h}}$$

The RHN does not bound its hidden state. Based on the criterion 5 in Section 3, we bound the hidden state using a similar structure as the GRU. We call the new structure Bounded Recurrent Highway Network (BRHN). The BRHN updates as follows:

$$\mathbf{r} = \sigma\left(\mathbf{W}_r\mathbf{x}_t + \mathbf{U}_r\mathbf{h}_{t-1}\right)$$
$$\tilde{\mathbf{h}} = \texttt{tanh}\left(\mathbf{W}\mathbf{x}_t + \mathbf{U}\mathbf{h}_{t-1}\right)$$
$$\mathbf{h}_t = \mathbf{r} \odot \mathbf{h}_{t-1} + (1 - \mathbf{r}) \odot \tilde{\mathbf{h}}$$

BRHN performs similarly to GATO and GRU on Penn TreeBank, while RHN numerically fails.

## H  EFFICIENT UNITARY RECURRENT NEURAL NETWORK (EURNN)

Efficient Unitary Recurrent Neural Network (EURNN) (Jing et al., 2017) is a fast and tunable version of the Unitary Recurrent Network (Arjovsky et al., 2016). The forward propagation in EURNN follows the RNN framework:

$$\mathbf{h}_t = f(\mathbf{W}\mathbf{x}_t + \mathbf{U}\mathbf{h}_{t-1})$$

Define $D$ to be diagonal matrix and $F_1, \ldots, F_L$ to be a sequence of rotation matrices. EURNN uses a parameterization to make $\mathbf{U}$ unitary:

$$\mathbf{U} = \mathbf{D}\mathbf{F}_1 \ldots \mathbf{F}_L$$

If $L$ equals to the number of rows of $\mathbf{U}$, it can capture the full space of the unitary matrices. Following Jing et al. (2017), we choose $L = 2$ to have faster computation and use the same nonlinearity

$$f(z_i) = \frac{z_i}{|z_i|} * \texttt{ReLU}(|z_i| + b_i),$$

where $b_i$ is a trainable parameter. Though EURNN has bounded hidden-to-hidden gradients, we know from Section 3 that these are not the only gradients that matter. The forward propagation of the hidden state can blow up, which may still cause gradient problems. It is a good research question whether it is possible to have both unitary hidden-to-hidden transformations and a bounded hidden state.

# I STATISTICAL RECURRENT UNIT (SRU)

Statistical Recurrent Unit (SRU) (Oliva et al., 2017) uses a moving average update for hidden state $\boldsymbol{\mu}_t$. In the update of each time step $t$, SRU first summarizes the information from the previous hidden state $\boldsymbol{\mu}_{t-1}$ into $\mathbf{r}_t$ and then calculates the candidate hidden state $\boldsymbol{\phi}_t$ based on $\mathbf{r}_t$ and the inputs $\mathbf{x}_t$. The final update is a weighted average of previous hidden state $\boldsymbol{\mu}_{t-1}$ and candidate $\boldsymbol{\phi}_t$. SRU has separate hidden states $\{\boldsymbol{\mu}_t^{(\alpha)}\}, \alpha \in \boldsymbol{\alpha}$. Following Oliva et al. (2017), we choose $\boldsymbol{\alpha} = \{0, 0.25, 0.5, 0.9, 0.99\}$ in our experiments. The SRU updates as follows:

$$\mathbf{r}_t = \text{ReLU}(\mathbf{W}_r \boldsymbol{\mu}_{t-1})$$
$$\boldsymbol{\phi}_t = \text{ReLU}(\mathbf{W}_\phi \mathbf{r}_t + \mathbf{W}_x \mathbf{x}_t)$$
$$\boldsymbol{\mu}_t^{(\alpha)} = \alpha \boldsymbol{\mu}_{t-1}^{(\alpha)} + (1 - \alpha)\boldsymbol{\phi}_t, \quad \forall \alpha \in \boldsymbol{\alpha}.$$

If one $\alpha$ equals to one, the corresponding hidden state $\boldsymbol{\mu}_t^{(1)}$ has the identity hidden-to-hidden gradient matrix property. However, when $\alpha = 1$, $\boldsymbol{\mu}_t^{(1)} \equiv \boldsymbol{\mu}_{t-1}^{(1)}$ will never be updated. To capture long-term dependencies in sequence modeling, we want changes in the hidden state earlier in time to affect the later hidden states. If $\boldsymbol{\mu}_t^{(1)}$ never changes, then we cannot use it to model long-term dependencies. In GATO, the update of $\mathbf{s}_t$ does not depend on $\mathbf{s}_{t-1}$ but is updated based on $\mathbf{r}_{t-1}$ and $\mathbf{x}_t$. Therefore, GATO not only has the identity hidden-to-hidden gradient matrix property for $\mathbf{s}_t$ but also can use $\mathbf{s}_t$ to model long-term dependencies.

For $\mu_t^{(\alpha)}, \alpha \neq 1$, the update of $\boldsymbol{\mu}_t^{(\alpha)}$ depends on $\boldsymbol{\mu}_{t-1}^{(\alpha)}$. Therefore, when $\alpha$ does not equal to 1, SRU does not have the same hidden-to-hidden identity matrix gradient property as GATO.

# J LSTMS ALSO HAVE GRADIENT PROBLEMS

By the form of the update $\mathbf{c}_t = \mathbf{f}_t \odot \mathbf{c}_{t-1} + \mathbf{i}_t \odot \mathbf{g}_t$ in Equation (7), $\partial \mathbf{c}_T / \partial \mathbf{c}_t$ has two gradient paths: one through each term in the sum. The first path is only linear in the cell state. This has led to claims that the LSTM avoids vanishing gradients by allowing information to flow along cell states over long time intervals. However, further analysis reveals that the second path depends on the entries of the weight matrices $\mathbf{W}_{gh}, \mathbf{W}_{fh}, \mathbf{W}_{ih}, \mathbf{W}_{oh}$, exponentiating their magnitude to power $T - t$ (Kanuparthi et al., 2019). The magnitude of the second gradient path can suppress the first so that weight updates are made in favor of short-term dependencies.

