# OpenReview forum: "GATO: Gates Are Not the Only Option"
_ICLR.cc/2020/Conference — Reject_

### Official Review · AnonReviewer2 · 2019-10-16
**Official Blind Review #2**

**Rating:** 8

**Review:**

The paper proposes a new RNN architecture designed to overcome vanishing/exploding gradient problems and to improve long-term memory for sequence modelling. The main ideas are (i) to split the hidden state into two parts, one of which does not influence the recurrence relation, and can therefore not blow up or contract by self-feedback; and (ii) to use periodic functions, in particular the cosine, as non-linearity in the decoder, so that the output is bounded but does not saturate.

The paper puts forward a fairly systematic analysis of the gradients in RNNs. The analysis appears correct, and is in fact quite similar to considerations in earlier RNN work (which is correctly cited), and forms the basis for the proposed GATO unit. There are two loose ends in this part:
1) the cosine non-linearity results from a purely negative selection - the function should be bounded, but not saturating. The paper does not even ask the question which periodic function might be a good choice.
2) While the method is presented as a grand theory, with the only constraint that a part of the hidden state does not influence the recurrence function; the actual implementation and experiments are limited to the narrow special case of "non-interacting" GATO, where the "passive" variables make up exactly half of the hidden vector, and the update of each individual hidden variable is influenced only by a single variable from the previous state. So there are in fact no empirical results, not even on toy data, for the general case that the paper claims to introduce.

In the experiments, there are two artificial problems (copying, adding) for sequences of symbols. These are illustrative and sensible to verify and analyse  the behaviour of GATO in a controlled setting, but rather far from most real sequence modelling tasks. In a third experiment the task is to classify whether or not patients will stay in intensive care for >1 week, based on a (seemingly private) database of time series of vital parameters. Unfortunately, the results for that experiment do not go beyond the usual "ours is better". While the numbers clearly support GATO, it would have been nice to look a bit closer and pinpoint what makes the difference. Also, the comparison is a bit loose. It would have been better to add additional baselines where also LSTM and GRU are restricted to "non-interacting", element-wise recurrence (as far as technically feasible). As it stands, it is unfair to claim "we can do it with much fewer parameters" - perhaps LSTM / GRU could, too. In fact, it could even be that the task is just simple, so that more restricted model with fewer parameters generally perform better - I do not claim this is the case, but the experiments do not rule it out and, hence, do not confirm that the clever GATO recurrence makes the difference.

A small gap is also that even the "real" experiment might not be completely realistic. Nowadays it is a popular strategy to use deep LSTMS / GRUs, i.e., stack multiple levels of recurrence, possibly with temporal sub-sampling, to better capture long-term relations. While this is potentially even more brittle, because of the additional gradient flow across layers, it does seem to work. But deep RNNs are not tested (in fact, not even mentioned) in the paper.

Overall, in spite of a few loose ends, I find both the design and the practical performance of the proposed GATO unit very interesting, and potentially valuable for the ICLR crowd. This is one of the more convincing RNN papers I have recently read.

**Experience Assessment:**

I have published one or two papers in this area.

**Review Assessment: Checking Correctness Of Derivations And Theory:**

I assessed the sensibility of the derivations and theory.

**Review Assessment: Checking Correctness Of Experiments:**

I assessed the sensibility of the experiments.

**Review Assessment: Thoroughness In Paper Reading:**

I read the paper at least twice and used my best judgement in assessing the paper.

---

> ### Author Response · Authors · 2019-11-14
> **Thank You For Your Feedback**
>
> >[Explore other periodic functions for decoder]
>
> In new Appendix B, we observe that sin has similar performance. We believe our criteria are necessary, but not necessarily sufficient. We believe finite sums of cos/sin may work well too.
>
> >[Experiment with different proportion of “passive” variables in GATO]
>
> In revised Appendix A, we explore using no passive variables and 1/4 instead of 1/2 of the state. Our findings are that Add and Penn TreeBank suffer with less passive variables. This suggests that having a larger fraction of the state devoted to long-term gradient propagation is important.
>
> Exploring fully-interacting GATO is a worthwhile direction.  We believe it is not necessary for the Adding and Copying tasks. For the real data tasks, it would be interesting to study a more powerful model (with interaction) combined with regularization (like recurrent dropout).
>
> >[MIMIC is a seemingly private database of vitals]
>
> Sorry for the incomplete information. MIMIC-III is a publicly accessible critical care database.
>
> >[Results do not go beyond “ours is better”. Look more closely at what makes the difference]
>
> In addition to better accuracy and perplexity, we emphasize stability across learning rates and seeds. As mentioned, we added new experiments on varying GATO’s non-linearities and proportion of “passive” variables.
>
> >[Compare against GRU/LSTM with diagonal weight matrices]
>
> Thank you for this suggestion. As mentioned, we have added diagonal LSTM and GRU, in one case matching hidden size and in another matching number of parameters. GATO outperforms all GRU and LSTM variants on our experiments.
>
>
> >[Unfair claims about fewer parameters, compare against models with similar parameter counts]
>
> This is a great point. As mentioned, we included comparisons against GRU/LSTM where we balanced the parameter counts. We have also removed excess emphasis on fewer parameters on the tasks where held-out metrics are reported.
>
> Theory suggests that optimization is easier in the overparameterized setting. See Arora et al. 2018.
>
> Arora et al. On the Optimization of Deep Networks: Implicit Acceleration by Overparameterization. 2018
>
> >[ Multi-layer RNNs are not compared or mentioned in this work ]
>
> We follow other recent RNN papers (SRU, FRU, EURNN) by focusing on fundamental design choices and basic tasks rather than explore the full range of deep variants or regularization.
>
> We have added details that some LSTM papers we cited use multiple layers, and mentioned deep variants in our conclusion.

---

### Official Review · AnonReviewer3 · 2019-10-23
**Official Blind Review #3**

**Rating:** 3

**Review:**

In this paper, the authors propose a novel recurrent architecture called GATO.
Specifically, the authors focused on sequence modeling tasks and developed criteria for RNN models on such tasks.
Tha GATO model can resolve the vanishing/exploding gradient issue and is robust to initializations.
Empirical results show GATO can outperform LSTM and RNN in both synthetic datasets and real datasets.

The key insight of the proposed model is that only part of the hidden states is recurrently updated.
The GATO achieves this by adding the skip connection channel (or residual connection) along the temporal dimension.
GATO summarizes the hidden states r_t by recurrently adding (transformed) r_t to s_t.
This idea also appears in many previous RNN models, such as highway RNN/LSTM, Statistical/Fourier Recurrent Units [1][2].
Specifically, the proposed GATO is a special case of SRU ( alpha=1). This limits the novelty of the paper and thus make the contribution marginal.

As for the experimental studies, the authors only provide comparisons with LSTM and GRU. There are a lot of advanced RNN architectures to address vanishing/exploding gradient issues, such as uRNN[3], oRNN[4], Spectral-RNN[5] and SRU/FRU [1][2]. It would be more convincing if the
authors could include these models into comparison.

Overall I think this paper should be further improved before being accepted.



[1] Oliva, J.B., Póczos, B. and Schneider, J., The statistical recurrent unit.
In ICML 2017 (pp. 2671-2680).

[2] Zhang, J., Lin, Y., Song, Z. and Dhillon, I., Learning Long Term Dependencies via Fourier Recurrent Units.
In ICML 2018 (pp. 5810-5818).

[3] Arjovsky, M., Shah, A. and Bengio, Y., Unitary evolution recurrent neural networks.
In ICML 2016 (pp. 1120-1128).

[4] Mhammedi, Z., Hellicar, A., Rahman, A. and Bailey, J., Efficient orthogonal parametrisation of recurrent neural networks using householder reflections.
In ICML 2017 (pp. 2401-2409).

[5] Zhang, J., Lei, Q. and Dhillon, I., Stabilizing Gradients for Deep Neural Networks via Efficient SVD Parameterization.
In ICML 2018 (pp. 5801-5809).

**Experience Assessment:**

I have published in this field for several years.

**Review Assessment: Checking Correctness Of Derivations And Theory:**

I carefully checked the derivations and theory.

**Review Assessment: Checking Correctness Of Experiments:**

I assessed the sensibility of the experiments.

**Review Assessment: Thoroughness In Paper Reading:**

I read the paper thoroughly.

---

> ### Author Response · Authors · 2019-11-14
> **Thank You For Your Feedback**
>
> > [Difference between GATO, SRU/FRU, and RHN]
>
> Thank you for these references.
>
> The difference between SRU with alpha=1 and GATO with respect to the identity Jacobian is subtle. In SRU, if alpha^j = 1, then the hidden state mu^j for that alpha has an identity Jacobian [d mu^j_t / d mu^j_t-1]. But in this case, mu^j stays constant for all t and cannot be used to capture long-term dependencies. When alpha^j does not equal 1, the Jacobian is not an identity matrix. In GATO, the hidden state is broken in h=[r,s]. Though [ds_t / ds_t-1] is an identity matrix, s is updated based on r and x and can be used to capture long term dependencies. Therefore GATO is not a special case of SRU. We have added this discussion to new Appendix Section I.
>
> FRU is a follow-up work to SRU. Each hidden state h_t in FRU depends on h_{t-1} so it does not have the identity matrix Jacobian.
>
> GATO is also different from Recurrent Highway Network. The s_t in GATO does use a skip-connection/residual structure. But the residual part depends only on r_{t-1} and x_t, not on s_{t-1}. This novel residual update renders the identity matrix Jacobian. The highway network does not have this special residual update.
>
> >[Compare with other models that address vanishing gradients such as URNN and SRU]
>
> Thanks for the suggestion. We have included comparisons against other recently proposed RNNs that address the vanishing gradient issue: RHN, EURNN (Jing, 2017), and SRU. RHN and EURNN perform poorly and NAN because they have unbounded forward propagation. SRU performs well on MIMIC but not on other tasks.
>
> We fix RHN’s performance with one principle from our paper. See the Penn TreeBank result.

---

### Official Review · AnonReviewer1 · 2019-10-23
**Official Blind Review #1**

**Rating:** 3

**Review:**

This paper proposes a modification of RNN that does not suffer from vanishing and exploding gradient problems. The proposed model, GATO partitions the RNN hidden state into two channels, and both are updated by the previous state. This model ensures that the state in one of the parts is time-independent by using residual connections. The experiments on the long copy and adding tasks, as well as language modeling on the Penn TreeBank dataset show the performance improvement against the basic LSTM and GRU.

The paper tackles an interesting and challenging problem with a novel approach in sequence modeling. The idea is clear and the paper is well-written. The mathematical insights are well reasoned.

The proposed method outperforms LSTM and RNN with much fewer number of parameters. However, there is no regularization is used for such big LSTM/GRU models. There is a chance that such a big LSTM/GRU model increased the chance of overfitting, and therefore the performance is low. I would like to see the comparison after adding any common regularization that prevents overfitting across the recurrent connections.

There are many advanced RNN/LSTMs proposed in recent years [1-5] addressing the vanishing gradient problem. It is hard to judge the quality of the proposed method due to the lack of evaluation/comparisons. This paper needs more intensive evaluations with recent RNN-based methods. For instance based on [6], AWD-LSTM [6] and RHN [4] achieved 52.8 and 65.4 test perplexity scores on the Penn TreeBank dataset respectively. The the best score in this paper is 112.85.

[1] "Phased LSTM: Accelerating recurrent network training for long or event-based sequences." 2016.
[2] "Fast-slow recurrent neural networks." 2017.
[3] "Skip RNN: Learning to skip state updates in recurrent neural networks." 2017.
[4] "Recurrent highway networks." 2017.
[5] "Dilated recurrent neural networks." 2017.
[6] "Regularizing and optimizing LSTM language models." 2017

All experiments are performed with 1 or 2 layers. Hierarchical RNN/LSTM performs much better in sequence learning. Is there any reason authors only showed 1 or 2 layers? How does GATO change with more than 2 layers?

How do other choices of non-linear functions affect the performance in practice?


Typo
r -> r_t in Eq. 6

---
After rebuttal:
One of my main concerns, weak baselines and unfair comparisons, was partially answered in the updated paper.  I am not fully convinced by their new comparisons.
For instance, authors mentioned that 'RHN and EURNN performed poorly because they have unbounded forward propagation'. To overcome this, they introduced 'Bounded RHN' in Append G and it performs similarly to GATO and GRU. However, this 'Bounded RHN' is the one used in the original RHN paper. Overall, it is hard to trust their additional comparisons.
Although, I believe that this paper is well-structured and justified. Also it has high potential for the community. However, the paper itself is not ready to be published.


**Experience Assessment:**

I have published in this field for several years.

**Review Assessment: Checking Correctness Of Derivations And Theory:**

I assessed the sensibility of the derivations and theory.

**Review Assessment: Checking Correctness Of Experiments:**

I carefully checked the experiments.

**Review Assessment: Thoroughness In Paper Reading:**

I read the paper thoroughly.

---

> ### Author Response · Authors · 2019-11-14
> **Thanks For Your Feedback**
>
> >[Large LSTM/GRU with no regularization may have overfit]
>
> Thank you for your comments.
>
> Our goal for this work is not to investigate generalization but rather to design sequence models that capture long-term dependencies.
>
> On Copy and Add, the models are trained on new samples from the data distribution at each batch. There is no notion of overfitting on these experiments.
>
> For MIMIC and Penn, we report held-out accuracy and perplexity.
>
> We have added comparisons against GRUs/LSTMs with similar parameter counts as GATO. GATO still outperforms these models on generalization in this non-regularized setting.
>
> Years of research has been devoted to regularizing GRUs/LSTMs. A future direction is to see whether the same techniques apply to alternate models such as GATO and RHN.
>
> >[Clarify why the lowest perplexity score in this paper is higher than those in other recent works]
>
> The best held-out perplexity score on Penn TreeBank of 112.85 is for unregularized models with no dropout. [4] and [6] achieve 65.4 and 52.8 using many training and regularization techniques.
>
> RNN Regularization (Zaremba, 2015) reports 114.5 test perplexity for this task for an unregularized LSTM.
>
> We have included comparisons against other recently proposed RNNs that address the vanishing gradient issue: RHN [4], EURNN (Jing, 2017), and SRU (Oliva, 2017). RHN and EURNN perform poorly and NAN because they have unbounded forward propagation. SRU performs well on MIMIC but not on other tasks.
>
> We fix RHN’s performance with one principle from our paper. See the Penn TreeBank result.
>
> >[GATO used 1 or 2 layers. How does GATO change with >2 layers?]
>
> Our “two layer” variant of GATO was not named clearly. “Two layer” for GATO referred to increasing the depth of the function used to compute the recurrence. This is different from using N stacked identical RNNs for N layers. The “two layers” in GATO is more similar to how the GRU takes several functions of the input state.
>
> We follow other recent RNN papers (SRU, FRU, EURNN) by focusing on fundamental design choices and basic tasks rather than explore the full range of deep variants or regularization.
>
> >[Other non-linear functions?]
>
> In new section Appendix B, we replace the sigmoid in the GATO update with tanh, and we replace the decoder cos with sin. We observe little to no variation.
>
> >[typo r -> r_t in Eq. 6]
>
> We fixed this typo. Thank you.

---

### Author Response · Authors · 2019-11-14
**Updated Paper and Response to Reviews**

We thank all of the reviewers and the AC for their time and feedback. The reviewers’ responses are positive in general.

The main contribution of this work is a list of criteria necessary for sequence models to capture long-term dependencies when trained with gradient-based optimization, along with one instantiation of a model that meets these criteria.

Reviewer #2 finds “both the design and the practical performance of the proposed GATO unit very interesting, and potentially valuable for the ICLR crowd.” and mentions “This is one of the more convincing RNN papers I have recently read.”

All reviewers suggested additional experiments. Our added experiments include 3 recent alternative RNN models and LSTM/GRU variants.

The reviewers asked for ablations on the partition of GATO’s hidden state and choice of non-linear functions. We have added this to Appendices A and B.

Two points of clarification:

1. We follow other recent RNN papers (SRU, FRU, EURNN) by focusing on fundamental design choices and basic tasks rather than explore the full range of deep variants or regularization. A future direction is to see whether the GRU/LSTM regularization techniques apply to our model or its deep variants.

2.GATO is not a special case of the SRU as mentioned by Reviewer #3. Additional Discussion in Appendix I

We uploaded a new version of the paper that addresses all of the reviewers' concerns. We cite the papers they mentioned.

Our model competes with or outperforms GRU, LSTM, and other recent RNNs. Our model exhibits stability in cases where the alternatives are unstable. Ablations do not indicate sensitivity of GATO to small choices of non-linearities.

---

### Author Response · Authors · 2019-11-15
**Plots in PDF Rendering Slowly in Browser**

Hello Reviewers and AC,

We have noticed that our submission's plots render slowly when viewing the PDF in some browsers. For us, it views OK in Mac Preview, but is slow on Chrome browser.

Apologies for the trouble. Thank you!

---

### Decision · Program_Chairs · 2019-12-19

**Decision:**

Reject

**Comment:**

This paper proposes a modification of RNN that does not suffer from vanishing and exploding gradient problems. The proposed model, GATO partitions the RNN hidden state into two channels, and both are updated by the previous state. This model ensures that the state in one of the parts is time-independent by using residual connections.

The reviews are mixed for this paper, but the general consensus was that the experiments could be better (baseline comparisons could have been fairer). The reviewers have low confidence in the revised/updated results. Moreover, it remains unclear what the critical components are that make things work. It would be great to read a paper and understand why something works and not that something works.

Overall: Nice idea, but the paper is not quite ready yet.